# Periodontal Regenerative Therapy Using rhFGF-2 and Deproteinized Bovine Bone Mineral versus rhFGF-2 Alone: 4-Year Extended Follow-Up of a Randomized Controlled Trial

**DOI:** 10.3390/biom12111682

**Published:** 2022-11-12

**Authors:** Fumi Seshima, Takahiro Bizenjima, Hideto Aoki, Kentaro Imamura, Daichi Kita, Daisuke Irokawa, Daisuke Matsugami, Yurie Kitamura, Keiko Yamashita, Hiroki Sugito, Sachiyo Tomita, Atsushi Saito

**Affiliations:** 1Department of Periodontology, Tokyo Dental College, Tokyo 1010061, Japan; 2Chiba Dental Center, Tokyo Dental College, Chiba 2618502, Japan; 3Oral Health Science Center, Tokyo Dental College, Tokyo 1010061, Japan; 4Department of Dental Hygiene, Tokyo Dental Junior College, Tokyo 1010061, Japan; 5Department of Operative Dentistry, Cariology and Pulp Biology, Tokyo Dental College, Tokyo 1010061, Japan

**Keywords:** periodontal bone loss, fibroblast growth factor 2, bone grafting, periodontitis, tissue regeneration, quality of life

## Abstract

The aim of this study was to evaluate longitudinal outcomes of recombinant human fibroblast growth factor (rhFGF)-2 plus deproteinized bovine bone mineral (DBBM) therapy in comparison with rhFGF-2 alone for treating periodontal intrabony defects. This study describes 4-year follow-up outcomes of the original randomized controlled trial. Intrabony defects in periodontitis patients were treated with rhFGF-2 (control) or rhFGF-2 plus DBBM (test). Clinical, radiographic, and patient-reported outcome (PRO) measures were used to evaluate the outcomes. Thirty-two sites were able to be followed up. At 4 years postoperatively, clinical attachment level (CAL) gains in the test and control groups were 3.5 ± 1.4 mm and 2.7 ± 1.4 mm, respectively, showing significant improvement from preoperative values but no difference between groups. Both groups showed an increase in radiographic bone fill (RBF) over time. At 4 years, the mean value for RBF in the test group (62%) was significantly greater than that in the control group (42%). In 1–2-wall defects, the test treatment yielded significantly greater RBF than the control treatment. No significant difference in PRO scores was noted between the groups. Although no significant difference in CAL gain was found between the groups at the 4-year follow-up, the combination treatment significantly enhanced RBF. Favorable clinical, radiographic outcomes, and PRO in both groups can be maintained for at least 4 years.

## 1. Introduction

In the concept of periodontal tissue engineering, a ‘biological agent’ is one of the critical elements needed [1]. Thus far, several such agents have been used clinically to regenerate periodontal tissues with certain degrees of success. Fibroblast growth factor (FGF)-2 induces strong angiogenic and proliferative activities in undifferentiated mesenchymal cells [2,3]. Cumulative evidence from basic research indicates that FGF-2 could stimulate cells in periodontal ligament toward regeneration [3]. The regenerative ability of FGF-2 was verified by subsequent clinical trials [4,5]. Since its approval in 2016, a commercial formulation of 0.3% recombinant human FGF-2 (rhFGF-2) has been widely used as the periodontal regenerative medicine in Japan [6,7].

Another important element in periodontal tissue engineering is the ‘scaffold’ [1]. In the regenerative treatment of periodontitis with residual intrabony pockets or certain bone defect configurations, the use of a biological agent with bone graft material is recommended [8]. A biological agent, enamel matrix derivative (EMD), has been used with various bone substitutes. A systematic review showed that EMD used with bone graft materials may yield additional gains in clinical attachment level (CAL) and reductions in probing depth, compared with the use of EMD alone [9]. In contrast, a randomized clinical trial (RCT) reported that the treatment of intrabony defects using EMD with alloplast or EMD alone showed comparable results after 36 months [10]. In the treatment of 2- and 3-wall defects, the combination therapy using EMD and synthetic bone graft failed to show any advantages over the use of EMD alone, after 4 years [11]. Thus, the true effects of such combination therapy are inconclusive.

Deproteinized bovine bone mineral (DBBM) is one of the most widely used bone graft materials in periodontal surgery [12,13]. Its use with a guided tissue regeneration membrane has been shown to be clinically effective [13,14,15,16,17]. It has been reported that the use of EMD with DBBM yielded greater improvements in clinical and radiographical outcomes compared with EMD alone, at 1 year following surgery [18]. Information, however, was lacking about the effectiveness of the use of DBBM with rhFGF-2 on periodontal regeneration.

Therefore, we conducted an RCT comparing the sole use of rhFGF-2 and rhFGF-2 plus DBBM in the treatment of intrabony periodontal defects [6]. We showed that both treatment modalities yielded similar values for CAL gain up to 2 years, but the combination treatment showed a significantly greater radiographic bone fill (RBF) [6,19]. Given the results from short-term studies, it is necessary to evaluate longitudinal outcomes of the combination treatment. In this extended follow-up study, we aimed to evaluate 4-year results of the sole use of rhFGF-2 versus rhFGF-2 plus DBBM as a means of treating intrabony defects.

## 2. Materials and Methods

### 2.1. Study Design

A 4-year extended follow-up of an RCT [6] was performed at Tokyo Dental College Suidobashi Hospital (Tokyo, Japan) and Tokyo Dental College Chiba Dental Center (Chiba, Japan). The original RCT was single-blind, randomized, controlled design, and evaluated healing at 6 months postoperatively. The study was approved by Tokyo Dental College Ethics Review Committee (No. 747) and conformed to the Declaration of Helsinki. The rationale, design, and the results obtained at the conclusion of the original study and at 2-year follow-up, have been published [6,19]. This study was registered at the University Hospital Medical Information Network-Clinical Trials Registry (UMIN-CTR) 000025257 and followed Consolidated Standards of Reporting Trials (CONSORT) guidelines.

### 2.2. Participants

In the original study, 32 moderate to severe chronic periodontitis [20] (in retrospect fulfilling the criteria for Stage III periodontitis [21]) patients were included [6]. A total of 44 defect sites were randomly divided into 2 groups (refer to the Appendix A flowchart).

For a more detailed report on the study participant inclusion and exclusion criteria, sample size calculation, randomization, and allocation procedures, please refer to the previously published paper [6]. In brief, the initial inclusion criteria were as follows: sites with probing pocket depth (PPD) >4 mm after initial periodontal therapy (IP) [22], with intrabony defect with depth of >3 mm, and an adequate plaque control level.

### 2.3. Clinical and Radiographic Examinations

Calibrated examiners assessed CAL, PPD, gingival recession (GR), bleeding on probing (BOP), and tooth mobility (TM) at post-IP, 6 months [6], 1 year, 2 years [19], and 4 years following surgery. Prior to the study, an investigator meeting was held. During the meeting, a calibration exercise in non-study volunteers was carried out. For PPD, all examiners reached the target SD of <0.4 mm. The computed kappa value for PPD ranged from 0.7 to 0.8. PPD was measured using a Williams type periodontal probe (YDM, Tokyo, Japan). PPD and GR were recorded in 0.5-mm increments. CAL was calculated as the sum of PPD and GR.

Periapical radiographs were taken using customized stents, and radiographic bone fill (RBF) (%) was analyzed [23].

### 2.4. Patient Reported Outcome (PRO) Measure

At each timepoint, the perception of oral health was rated by the participants, using an instrument for oral health-related quality of life (QoL)—the OHRQL-J [24,25].

### 2.5. Surgical Interventions

Mucoperiosteal flaps were elevated under local infiltration anesthesia. Following thorough degranulation of the intrabony site, scaling and root planing was performed. In the test group, 0.3% rhFGF-2 [REGROTH^®^ Dental Kit, 600 μg in hydroxypropyl cellulose (HPC); 100–200 µL, Kaken Pharmaceutical, Tokyo, Japan] with DBBM (Bio-Oss^®^, 0.25–1.0 mm granules, Geistlich Pharma AG, Wolhusen, Switzerland) was applied to the defect. Prior to the application, the rhFGF-2 solution was thoroughly mixed with DBBM in a sterile disposable dish. Attention was paid not to overfill the defect: approximately 80% to 90% of the defect was filled with the mixture. In the control group, only the rhFGF-2 formulation was given to the defect. Immediately after application, the flaps were repositioned for complete closure and sutured with modified vertical mattress or interrupted sutures.

### 2.6. Postsurgical Care

Patients received antimicrobials (amoxicillin 750 mg/day or cefdinir 300 mg/day) for 4 days. Standard pain medications were prescribed as needed. Patients used a mouthwash twice per day. They gently cleaned the operated area with an ultrasoft toothbrush, beginning 1 day postoperatively and continued for 4 weeks.

Sutures were removed after 10 to 14 days. The patients were then placed in supportive care programs.

### 2.7. Statistical Analysis

The CAL gain at 4 years following surgery served as the primary endpoint. The following statistical analyses were performed using InStat 3.10 or Prism 9.4.2 (GraphPad, San Diego, CA, USA). The difference between the two groups was analyzed using the Mann–Whitney U test. For the intragroup comparison over time, Friedman test with Dunn post test was used. Fisher’s exact test was used to analyze the categorical variables. Spearman rank correlation was used to analyze the correlation between preoperative parameters and CAL gain. To identify the baseline predictors of CAL gain at 4 years postoperatively (dependent variable), multiple regression analysis was employed. A *p* value of 0.05 was considered statistically significant.

## 3. Results

### 3.1. Participants

In the original study [6], 32 patients were enrolled. The study flowchart is shown in Appendix A. At the 4-year follow-up, 32 sites (16 in the test group (rhFGF-2 + DBBM) and 16 in the control group (rhFGF-2 only)) in 25 patients were analyzed (Appendix A). The reasons for dropout were accidental death, serious body injury, relocation with no forwarding address, and no shows.

The demographic information and clinical parameters at baseline of the participants in the current study is shown in Appendix A. Baseline data from the original 6-month study were reported previously [6].

### 3.2. Clinical Outcomes

Healing was uneventful in all participants. Representative clinical cases are shown in Figure 1. Table 1 shows baseline defect locations and configurations. No significant intergroup differences in defect position, morphology, depth, or width were found. 

In both groups, significant improvements in CAL and PPD from baseline values were found at all follow-up timepoints (Table 2). The extent of CAL improvement at 6 months has been retained for 4 years. At 4 years postoperatively, mean CAL gains in the test and control groups were 3.50 ± 1.41 mm and 2.72 ± 1.43 mm, respectively, showing no significant intergroup difference (Figure 2a).

At 4 years, 25.0% of sites (*n* = 4) in the test group showed CAL gains of >4 mm, whereas 18.8% (*n* = 3) in the control group yielded such outcome (Appendix A).

PPD reductions at 4 years in the test and control groups were 3.18 ± 1.55 mm and 3.25 ± 1.38 mm, respectively, with no significant intergroup difference.

### 3.3. Relationship between Baseline Variables and CAL Gain at 4 Years

Baseline CAL and PPD presented a significant positive correlation with CAL gain at 4 years in both groups (Appendix A). In the control group, the number of teeth had a positive correlation with CAL gain. In the test group, the baseline defect depth had a positive correlation with CAL gain.

In multiple regression analysis, no multicollinearity was found among the variables. No significant relationship was found between baseline variables and the CAL gain at 4 years in the control group (Appendix A). In the test group, baseline PPD and defect depth had significant associations with CAL gain at 4 years (Appendix A).

### 3.4. Radiographic Outcome

A trend for increase in RBF was observed in both groups (Figure 2b, Table 2). In the control group, the mean value for RBF at 4 years postoperatively was significantly greater than that at 6 months (Figure 2b). In the test group, the values for RBF at 4 years was significantly greater than those at 6 months and at 1 year.

At 4 years, RBF in the test group (61.8%) was significantly greater compared with the control group (41.5%) (Table 2).

### 3.5. CAL Gain and RBF in Different Defect Configurations

No significant difference in CAL gain was noted between 3-wall and 1–2-wall defects in either group (Table 3). In the control group, RBF was significantly greater in 3-wall defects than in 1–2-wall defects. In the test group, no such difference was found. In 1–2-wall defects, RBF in the test group was significantly greater than that in the control group.

### 3.6. OHRQL-J Scores

Compared with the mean total OHRQL-J score following IP, neither treatment yielded significant changes in scores over time (Figure 3). No significant intergroup difference in scores was found at any timepoint.

## 4. Discussion

Many RCTs are relatively short-term and, due to various reasons, they are seldom re-visited or extended [26]. Treatment effects can change beyond the initial or short-time follow-up study. Previously, we reported 2-year follow-up results [19] of the 6-month RCT [6], but 2-years is a relatively short follow-up period. There are currently no mid-term or long-term data available about the effectiveness of regenerative therapy employing rhFGF-2 and bone substitutes. Therefore, we thought it necessary to evaluate the longitudinal effects of the combined use of rhFGF-2 and DBBM on the healing of intrabony defects, as compared to rhFGF-2. The results of the current study showed that the mean CAL gains at 4 years postoperatively (primary endpoint) were 3.5 mm in the test group and 2.7 mm in the control group, with no significant intergroup differences. This finding is consistent with the results from our 2-year follow-up study [19] and studies comparing the use of EMD alone versus EMD plus bone substitutes [27,28]. The 4-year outcomes reported here suggest that rhFGF-2 alone and rhFGF-2 plus DBBM were comparably effective in the resolution of intrabony defects in the treated patients. Given that the rhFGF-2 formulation itself lacks space-making property, the favorable results from the use of rhFGF-2 alone (with no addition of bone substitutes or membranes) may further support the clinical efficacy of rhFGF-2 therapy.

It should be noted that Table 2 and Figure 2 indicate a trend for gradual CAL gain for the test group as the values in the control group appeared to be decreased during follow-up, although no statistically significant differences were found between the groups. The decreased sample size due to dropouts and the resulting reduction of statistical power may have concealed differences that were considered to be relevant.

In both groups, preoperative CAL and PPD were significantly positively correlated with CAL gains at 4 years postoperatively. In the multiple regression analysis in the test group, preoperative PPD and defect depth showed significant associations with CAL gains at 4 years. These data are consistent with the previous studies showing that greater CAL gain can be obtained after the regenerative therapy at the sites with greater pocket depth [29,30]. This also means that when comparing CAL or PPD data from various studies, care must be exercised.

When the individual outcome at 4 years was evaluated using the composite outcome measure (COM) for periodontal regeneration [31], 38% (*n* = 6) of the control group and 69% (*n* = 11) in the test group fulfilled the criteria of ‘successful’ treatment; CAL gain of >3 mm and residual PD (PPD) <4 mm. This may indicate the clinical advantage of the combination therapy, although no significant difference in the mean values of CAL gain at 4 years was found between groups. Considering the high variability of results among the participants, it may be beneficial to include COM as an evaluation method in future studies of periodontal regenerative therapy, because it presents a perspective different from just looking at the collective data.

Another crucial indicator of success for periodontal regenerative therapy is bone level. It was postulated that measures for hard and soft tissues should be combined to evaluate the clinical outcomes of a biological agent [32]. The test group’s mean RBF value at 4 years (56%) was significantly greater than the control group’s (41%). Due to the usage of radiopaque material with rhFGF-2, it can be claimed that the test sites should have a higher RBF value. It should be noted that RBF values in the test group also increased over time, suggesting the new bone formation, not merely observing the grafted material.

Clinically, it is important to know how different treatment approaches perform in different bone defect configurations. However, the prediction of treatment outcomes based on the characteristics of intrabony defects can be difficult [33]. In the present study, there was a significant positive correlation between preoperative defect depth and CAL gain at 4 years postoperatively in the test group. Moreover, the result of the multiple regression analysis indicated that the defect depth could predict the extent of CAL gain at 4 years. In the current study, no significant difference in CAL gain was found between 3-wall and 1–2-wall defects in both groups. As for the radiographic outcome, it is interesting to note that no significant difference in RBF of 3-wall defect was found between the groups. In 1–2-wall defects, however, the test treatment achieved significantly greater RBF than the control treatment. These data collectively suggest that the combination therapy is more effective than the sole use of rhFGF-2 in the resolution of deeper or poorly-contained periodontal defects. This finding is in line with the recommendation by Cortellini and Tonetti [34], who stated that the addition of bone substitutes may be needed in the regenerative treatment of non-contained defects. In an RCT of treatment of deep and non-contained intrabony defects with the use of EMD with DBBM, the clinical outcomes after 12 months were similar to the use of collagen barrier [35]. However, in the treatment of severe and poorly-contained bone defects, additional use of a barrier membrane may yield enhanced outcome, by providing wound stability and space needed for regeneration. This needs to be clarified by future studies.

The regenerative potential is influenced by the quantity and distribution of cell sources and vascularity in the area surrounding the defect [36]. In our recent pre-clinical experiment of the treatment of intrabony periodontal defects, significantly greater levels of proliferating cell nuclear antigen-, vascular endothelial growth factor-, and osterix-positive cells were observed in rhFGF-2 and rhFGF-2 + DBBM treated groups compared with the unfilled group [37]. In vitro, the addition of rhFGF-2 to DBBM promoted cell viability/proliferation, attachment/spreading, and osteogenic differentiation, as assessed by alkaline phosphatase and alizarin red staining [37]. The enhanced bone healing observed in the combination therapy group may be ascribed to the sustained release of FGF-2 from DBBM. We confirmed the release of FGF-2 from FGF-2-treated DBBM over 120 h [37]. This may explain the greater bone healing of 1–2-wall defects following the combination therapy used in the current study. Recently, Shirakata et al. reported that DBBM seems to be a suitable carrier for rhFGF-2 and that rhFGF-2/DBBM treatment promotes favorable periodontal regeneration compared with rhFGF-2, rhFGF-2/β-TCP, and rhFGF-2/CO_3_Ap treatments in one-wall intra-bony defects [38]. These findings collectively suggest that DBBM may be an effective scaffold or carrier to be used with rhFGF-2. Some studies reported no significant benefits when EMD was used in combination with bone graft materials [10,39]. Regarding the treatment of 1-wall defects, a 4-year follow-up of a RCT comparing the use of EMD alone versus EMD with demineralized porcine bone matrix showed no additional benefits for the combination therapy [40]. More research is needed to confirm the performances of different bone graft materials in rhFGF-2 therapy and to identify the indications.

There are limitations to this study. First, the dropout rate during the 4-year follow-up was relatively high. This reduced the statistical power, which could compromise the interpretation of the findings. Second, because of the limited sample size and design of the original study, it is difficult to draw definite conclusions regarding the treatment effect on different defect configurations. Finally, the follow-up period of this study was 4 years: a longer observation period is needed to properly evaluate the longitudinal stability of the treatment. Despite these limitations, this study provides novel information about the stability of the regenerative therapy using rhFGF-2 plus bone substitute.

## 5. Conclusions

This study is the first and so far the only one to provide prospective follow-up data on the combined use of rhFGF-2 with DBBM, in the treatment of intrabony defects. At 4 years postoperatively, the combination therapy yielded no significant additional benefit in terms of CAL gain but significantly enhanced radiographic bone level when compared with rhFGF-2 alone. In both treatment modalities, favorable clinical and radiographic outcomes and PRO can be preserved for at least 4 years.

## Figures and Tables

**Figure 1 biomolecules-12-01682-f001:**
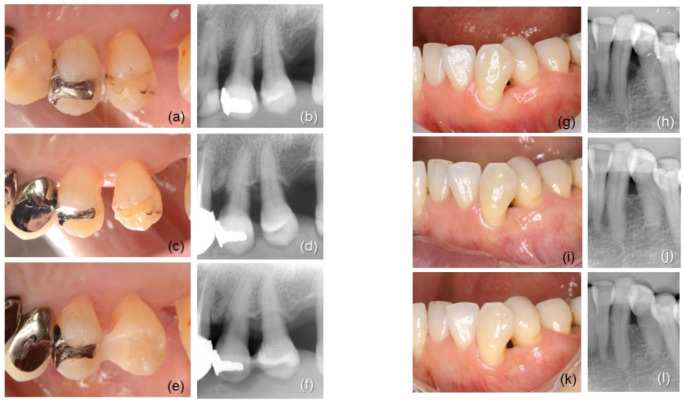
Representative clinical cases; (**a**–**f**) 60-year-old woman: intrabony defect in #24 was treated by rhFGF-2 + DBBM (test treatment); (**a**) preoperative view, the mesial site had PPD of 7 mm; (**b**) radiograph before surgery, defect width; 5 mm, depth; 3 mm; (**c**) 2-year follow-up; (**d**) radiograph at 2 years; (**e**) 4-year follow-up; PPD 2 mm; (**f**) radiograph at 4 years; (**g**–**l**) 53-year-old woman: #33 was treated by rhFGF-2 (control treatment); (**g**) preoperative view, PPD at the distal site was 7 mm; (**h**) radiograph before surgery, defect width; 3 mm, depth; 5 mm; (**i**) 2-year follow-up; (**j**) radiograph at 2 years; (**k**) 4-year follow-up; PPD 2 mm; (**l**) radiograph at 4 years.

**Figure 2 biomolecules-12-01682-f002:**
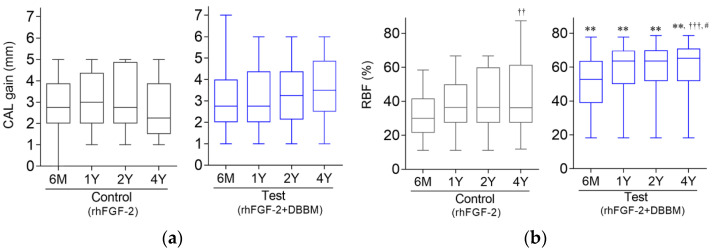
Clinical attachment level (CAL) gain (**a**) and radiographic bone fill (RBF) (**b**). The chart shows maximum, median, minimum, and 25th and 75th percentiles. ** *p* < 0.01, compared to the control group by Mann–Whitney U test. ^††^ *p* < 0.01, ^†††^ *p* < 0.001 compared to 6 months; # *p* < 0.05 compared to 1 year, Friedman test with Dunn post hoc test.

**Figure 3 biomolecules-12-01682-f003:**
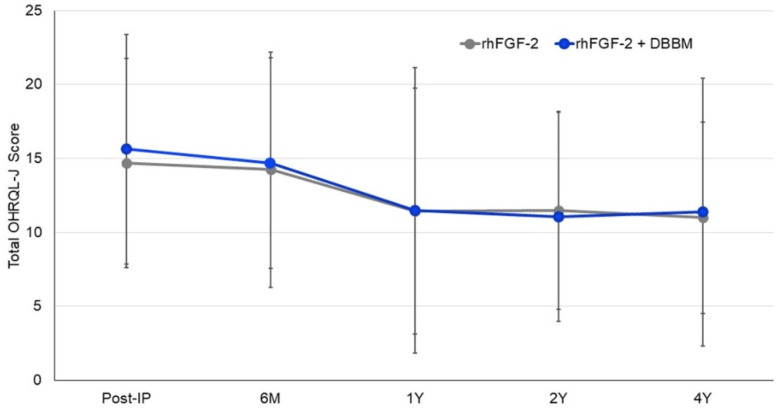
Change in total OHRQL-J scores. Data presented as mean ± standard deviation.

**Table 1 biomolecules-12-01682-t001:** Defect locations and configurations.

Intrabony Defect	rhFGF-2 (Control, *n* = 16)	rhFGF-2 + DBBM (Test, *n* = 16)
Position [*n* (%)]		
Maxilla	6 (37.5)	9 (45.0)
Mandible	10 (62.5)	7 (55.0)
Anterior teeth	5 (31.3)	2 (10.0)
Premolars	4 (25.0)	5 (25.0)
Molars	7 (43.7)	9 (65.0)
Morphology [*n* (%)]		
1-wall	3 (18.8)	2 (12.5)
2-wall	4 (25.0)	5 (31.3)
3-wall	4 (25.0)	4 (25.0)
combination	5 (31.2)	5 (31.3)
Depth (mm; mean ± SD)	4.72 ± 1.88 (range, 3.0–11.0)	4.53 ± 1.09 (range, 3.0–6.5)
Width (mm; mean ± SD)	2.88 ± 0.82 (range, 2.0–5.0)	3.59 ± 1.87 (range, 2.0–10.0)

**Table 2 biomolecules-12-01682-t002:** Clinical and radiographic outcomes of control (rhFGF-2) and test (rhFGF-2 + DBBM) groups (total *n* = 32 sites).

Variable/Group	Baseline (Post-IP)	6 Months	1 Year	2 Years	4 Years
**CAL** (mm)					
rhFGF-2	7.28 ± 1.73 (6.5; 6.36–8.20)	4.56 ± 1.44 ** (4; 3.80–5.33)	4.25 ± 1.53 *** (4; 3.44–5.06)	4.18 ± 1.35 *** (4; 3.47–4.90)	4.56 ± 1.25 ** (4.25; 3.90–5.23)
rhFGF-2 + DBBM	7.31 ± 1.62 (7; 6.45–8.18)	4.22 ± 1.17 *** (4; 3.60–4.84)	4.22 ± 0.97 ** (4.25; 3.70–4.73)	3.93 ± 0.89 *** (4; 3.46–4.41)	3.81 ± 0.91 *** (4; 3.33–4.30)
Difference between groups	N.S.	N.S.	N.S.	N.S.	N.S.
**PPD** (mm)					
rhFGF-2	6.28 ± 1.46 (5; 5.50–7.01)	2.94 ± 0.87 *** (3; 2.47–3.40)	2.75 ± 0.86 *** (3; 2.29–2.99)	2.68 ± 0.89 *** (2.25; 2.21–3.16)	3.03 ± 0.97 ** (2.75; 2.51–3.55)
rhFGF-2 + DBBM	6.00 ± 1.27 (6.5; 5.33–6.67)	2.66 ± 0.72 *** (2.50; 2.27–3.04)	2.72 ± 0.52 *** (3; 2.44–2.99)	2.59 ± 0.49 *** (3; 2.33–2.86)	2.63 ± 0.62 *** (2.50; 2.30–2.96)
Difference between groups	N.S.	N.S.	N.S.	N.S.	N.S.
**GR** (mm)					
rhFGF-2	0.93 ± 1.24 (0.50; 0.27–1.60)	1.37 ± 1.53 (1; 0.56–2.19)	1.50 ± 1.27 (1.5; 0.83–2.17)	1.50 ± 1.30 (1.25; 0.78–2.22)	1.53 ± 1.30 (1.25; 0.84–2.22)
rhFGF-2 + DBBM	1.31 ± 1.35 (1; 0.59–2.03)	1.56 ± 1.22 (1.25; 0.91–2.21)	1.44 ± 0.98 (1.25; 0.91–1,96)	1.25 ± 0.93 (1.25; 0.75–1.75)	1.22 ± 0.88 (1; 0.75–1.69)
Difference between groups	N.S.	N.S.	N.S.	N.S.	N.S.
**BOP** positive (%)					
rhFGF-2	62.5	12.5 ***	6.3 ***	0.0 ***	0.0 ***
rhFGF-2 + DBBM	62.5	6.3 ***	0.0 ***	0.0 ***	0.0 ***
Difference between groups ^a^	N.S.	N.S.	N.S.	N.S.	N.S.
**TM**					
rhFGF-2	0.11 ± 0.32 (0; −0.05–0.27)	0.06 ± 0.24 (0; −0.06–0.17)	0.06 ± 0.24 (0; −0.06–0.17)	0.11 ± 0.32 (0; −0.05–0.27)	0.11 ± 0.32 (0; −0.05–0.27)
rhFGF-2 + DBBM	0.20 ± 0.41 (0; 0.01–0.39)	0.05 ± 0.22 (0; −0.05–0.15)	0.05 ± 0.22 (0;−0.05–0.15)	0.05 ± 0.22 (0; −0.05–0.15)	0.05 ± 0.22 (0; −0.05–0.15)
Differencebetween groups	N.S.	N.S.	N.S.	N.S.	N.S.
**RBF** (%)					
rhFGF-2	–	31.2 ± 14.1 (28.7; 23.6–38.7)	36.7 ± 16.1 (34.9; 28.1–45.3)	39.6 ± 18.0 (34.9; 30.1–49.2)	41.5 ± 21.5 ^††^ (36.2; 29.6–53.4)
rhFGF-2 + DBBM	–	50.6 ± 16.7 (52.8; 41.7–59.5)	58.2 ± 15.7 (63.6; 49.9–66.6)	60.1 ± 15.8 (63.6; 51.7–68.5)	61.8 ± 16.0 ^†††^ (65.7; 53.2–70.3)
Difference between groups		*p* = 0.003	*p* = 0.001	*p* = 0.003	*p* = 0.006

Data shown as mean ± standard deviation (median; interquartile range), except for BOP. The difference between groups at each time point was assessed by the Mann–Whitney U test. Intragroup difference over time was assessed by the Friedman test with Dunn post test. ^a^ Categorical data were assessed by Fisher’s exact test. ** *p* < 0.01, *** *p* < 0.001, compared to baseline; ^††^ *p* < 0.01, ^†††^ *p* < 0.001, compared to 6M. IP, initial periodontal therapy; CAL, clinical attachment level; PPD, probing pocket depth; GR, gingival recession; BOP, bleeding on probing; TM, tooth mobility; RBF, radiographic bone fill.

**Table 3 biomolecules-12-01682-t003:** Comparison of clinical attachment level (CAL) gain and radiographic bone fill (RBF) at 4 years postoperatively between different defect configurations.

Outcome	Defect	rhFGF-2 (Control)	Difference	rhFGF-2 + DBBM (Test)	Difference
CAL gain (mm)	3-wall	3.19 ± 1.36 (3.50; 2.05–4.33)	N.S.	3.72 ± 1.52 (4.00; 2.55–4.89)	N.S.
	1–2-wall	2.25 ± 1.41 (2.00; 1.07–3.43)		3.21 ± 1.32 (3.00; 2.00–4.43)	
RBF (%)	3-wall	47.6 ± 14.4 (48.6; 36.6–58.7)	*p* = 0.03	61.4 ± 13.3 (65.2; 51.2–73.1)	N.S.
	1–2-wall	32.3 ± 28.1 (25.5; 2.7–61.8)		61.1 ± 19.6 * (67.3; 43.1–79.2)	

Data shown as mean ± standard deviation (median; interquartile range). Difference between different defect configurations within group or difference between groups within the same defect configuration was assessed by Mann–Whitney U test (* *p* = 0.032, compared to the control group).

## Data Availability

Supporting data for this study may be made available from the corresponding author upon reasonable request.

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
