# Peer review of "Periodontal Regenerative Therapy Using rhFGF-2 and Deproteinized Bovine Bone Mineral versus rhFGF-2 Alone: 4-Year Extended Follow-Up of a Randomized Controlled Trial"

_biomolecules, 2022, doi:10.3390/biom12111682_

Round 1
Reviewer 1 Report
Abstract. This section is correct and show the summary of the paper.
Introduction. This investigation is a paper that presents information for researchers in the field of regenerative therapy in periodontics. Fibroblast growth factor (FGF) -2 induces strong angiogenic and proliferative activities in undifferentiated mesenchymal cells. Cumulative evidence from basic research indicates that FGF-2 could stimulate cells in periodontal ligament toward regeneration.
The aim of this study was to evaluate longitudinal outcomes of recombinant human fibroblast growth factor (rhFGF)-2 plus deproteinized bovine bone mineral (DBBM) therapy in comparison with rhFGF-2 for treating periodontal intrabony defects.
This section is very short and must improved with more scientific evidence of backgrounf of the field.
Materials and methods.
This randomized controlled trial (RCT) was designed to compare the use of rhFGF-2 and rhFGF-2 plus DBBM in the treatment of intrabony periodontal defects.
The paper is a longitudinal study of a previous research. The initial inclusion criteria were as follows: sites with probing pocket depth (PPD) > 4 mm after initial periodontal therapy (IP), with intrabony defect with depth of > 3 mm, and an adequate plaque control level.
This section is correct and explain each step of methodology.
Results
In both groups, significant improvements in clinical attachment level (CAL) and PPD from baseline values were found at all follow-up timepoints. The extent of CAL improvement at 6 months has been retained during 4 years. At 4 years postoperatively, mean CAL gains in the test and control groups were 3.50 ± 1.41 mm and 2.72 ± 1.43 mm, respectively, showing no significant intergroup difference.
At 4 years, 25.0% of sites (n=4) in the test group showed CAL gains of > 4 mm, whereas 18.8% (n=3) in the control group yielded such outcome.
A trend for increase in radiographic bone fill was observed in both groups.
This section is correct and explain the clinical outcomes with figures and tables.
Discussion.
The authors discuss the results of the study. In fact, several paragraphs show the importance of clinical results of the paper. But the authors, must include a greater discussion of results with other studies about the regeneration of periodontal bone defects, according to the scientific evidence.
Conclusively, the study is not ready for publication.
Author Response
We are grateful for your time and effort in reviewing our manuscript and for the helpful comments and advice.
Introduction.
This section is very short and must improved with more scientific evidence of background of the field.
(Response)
Thank you for this comment. We added more background information regarding the clinical results of the combination therapy.
Materials and methods.
This section is correct and explain each step of methodology.
(Response)
We thank the reviewer for the positive comment.
Results
This section is correct and explain the clinical outcomes with figures and tables.
(Response)
Thank you for the encouraging comment.
Discussion.
The authors discuss the results of the study. In fact, several paragraphs show the importance of clinical results of the paper.
(Response)
We appreciate the reviewer for the positive comment.
But the authors, must include a greater discussion of results with other studies about the regeneration of periodontal bone defects, according to the scientific evidence.
(Response)
This study is the extended follow-up study of the previous 6 month-RCT and 2-year follow-up study. And there have been no other studies comparing the use of rhFGF-2 and rhFGF-2+DBBM. Given these, it is difficult to discuss in relation to other studies on periodontal regeneration. However, in light of the reviewer’s advice, we added such discussion in our revised manuscript.
Conclusively, the study is not ready for publication.
(Response)
We extensively revised our manuscript, according to the suggestions by the reviewers.
Reviewer 2 Report
The results of a follow-up study of 4 years through clinical research are considered to be important results.
The description of the functional mechanism for the combined use of rhFGF-2 and DBBM in the treatment of intraosseous defects is somewhat lacking, and further explanation is needed.
I think it should be written in detail about the dosage of rhFGF-2 and DBBM used during the treatment procedure.
It is judged that the treatment may be different depending on the dosage of the commercially available formulations.
Author Response
We are grateful for your time and effort in reviewing our manuscript and for the helpful comments and advice.
The results of a follow-up study of 4 years through clinical research are considered to be important results.
(Response)
We thank the reviewer for such a positive comment.
The description of the functional mechanism for the combined use of rhFGF-2 and DBBM in the treatment of intraosseous defects is somewhat lacking, and further explanation is needed.
(Response)
We understand the reviewer’s view. However, please understand that this study is the extended follow-up of the previous 6-month RCT and 2-year follow-up study. Thus, the clarification of the functional mechanism for the combined use is not the purpose of this study. However, in light of the reviewer’s comment, we added more information from our basic study to better discuss functional mechanisms for the combined use.
I think it should be written in detail about the dosage of rhFGF-2 and DBBM used during the treatment procedure. It is judged that the treatment may be different depending on the dosage of the commercially available formulations.
(Response)
We thank the reviewer for the important comment. We added more information on the dosage of rhFGF-2 and DBBM. As the reviewer pointed out, it is possible that different dosage of the formulations may have exerted different effects on the outcome. However, we used the same concentration (0.3%) of the commercial formulation of rhFGF-2. And the amount of DBBM needed to be used in relation to the volume of the existing defects. Therefore, the greater the defect, the greater amount (volume) of rhFGF-2 and DBBM was used. Considering this, it may not be the case that the greater the amount of the formulations, the greater the levels of outcomes.
Reviewer 3 Report
Title: Periodontal Regenerative Therapy using rhFGF-2 and Deprotein-2 ized Bovine Bone Mineral versus rhFGF-2 alone: 4-year Extended 3 Follow-up of a Randomized Controlled Trial.
The authors have already published two articles with the same objectives and due to the way they have written the current one, they force the reader to have to read the previous ones as well. This is not correct. The manuscript lacks a clear justification for conducting this new study.
Other methodological errors and shortcomings are described below.
KEY WORDS:
- Use appropriate KeyWords MeSH terms so that if the manuscript is accepted, it can be located in the bibliographic repertoires (PubMed).
M&M
-RCTs must compulsorily meet two methodological requirements: compliance with a CONSORT checklist and registration on the website http://clinicaltrials.gov
-The authors state: ...." In brief, the initial inclusion criteria were as follows: sites with probing pocket depth (PPD) > 4 mm after initial periodontal therapy...". In Stage III of periodontitis, the average PPD in patients is ≥6mm and ≥5mm of CAL. How were periodontal pockets >4mm selected? Would the patients be Stage I or II?. Clarify.
-The authors state that "calibrated examiners assessed" up to 5 periodontal variables. Don't the authors believe that measures of intra- and inter-examiner concordance agreement should be described?.
-With which periodontal probe did the authors conduct the study?.
-As the authors describe in the penultimate paragraph of the introduction, they justify the wide use of DBBM in periodontal regeneration, with GTR techniques. Why have the authors not used membranes as a barrier mechanism in their essay, and a fundamental requirement of these techniques?.
- How was the regression model built, what were the independent variables?.
RESULTS
-The authors continually refer to the two previous published studies, this creates confusion. If the present, I include 25 patients (versus 32) and 32 analyzed sites, (versus 44), therefore if they intend to consider different studies, the authors cannot direct the readers to the previous ones for the demographic information and clinical parameters. The number of patients and the number of sites have changed and readers need to know these facts.
-The authors should under discussion highlight the differences with the two studies already published (at 6 months and at two years) and the justification for evaluating the same variables at 4 years. What does this new assessment contribute? As can be seen in the tables and graphs, most of the clinical and OHRQL variables are identical to those obtained at two years. Only RBF in comparison between groups is different, and when it is compared with the results at 6 months, 1 year, and 2 years, table 2? Why don't the authors provide this information to demonstrate or justify this study?.
DISCUSSION
-What does it mean: ….Given that the rhFGF-2 formulation itself lacks space-making property, this finding may further support the clinical efficacy of rhFGF-2 therapy?. As a reader, I take it that since rhFGF-2 is a non-space-creating substance, and since it does not create additional benefits along with DBBM, it would be more logical to use the latter product, which is safe and highly accepted as regenerative.
Author Response
We are grateful for your time and effort in reviewing our manuscript and for the helpful comments and advice.
The authors have already published two articles with the same objectives and due to the way they have written the current one, they force the reader to have to read the previous ones as well. This is not correct.
(Response)
As the reviewer pointed out, this is the extended follow-up study of the 6-month RCT and the 2-year follow-up study. Given the nature of the follow-up study, it is necessary to draw readers’ attention to our previous publication. However, we completely understand the reviewer’s concern. Therefore, we added descriptions to the revised manuscript.
The manuscript lacks a clear justification for conducting this new study.
(Response)
Most RCTs are relatively short term and, due to various reasons, they are seldom re-visited or extended (Davies et al. Scientific Reports, 8(1), 1-8, 2018). Obviously, there is no guarantee that treatment effects remain unchanged beyond the initial or short-time follow-up study. We feel that this study adds to the existing literature by providing longitudinal data on the outcomes of the use of rhFGF-2 with DBBM.
KEY WORDS:
- Use appropriate KeyWords MeSH terms so that if the manuscript is accepted, it can be located in the bibliographic repertoires (PubMed).
(Response)
We thank the reviewer for the important advice. We made changes accordingly.
M&M
-RCTs must compulsorily meet two methodological requirements: compliance with a CONSORT checklist and registration on the website http://clinicaltrials.gov
(Response)
This study has been registered on the UMIN registry. CONSORT information was added to the revised manuscript.
-The authors state: ...." In brief, the initial inclusion criteria were as follows: sites with probing pocket depth (PPD) > 4 mm after initial periodontal therapy...". In Stage III of periodontitis, the average PPD in patients is ≥6mm and ≥5mm of CAL. How were periodontal pockets >4mm selected? Would the patients be Stage I or II?. Clarify.
(Response)
We selected sites with PPD > 4 mm, according to the criteria of the Japanese Society of Periodontology 2015. We added the reference to the revised manuscript. The reviewer’s comment “In Stage III of periodontitis, the average PPD in patients is ≥6mm and ≥5mm of CAL” is not correct. Criteria for Stage III periodontitis include “interdental at the site of greatest loss > 5mm (not the average CAL). And the “complexity” criterion is “maximal PD > 6 mm.” Because we made patent-based diagnosis (not site basis), the target sites were not necessary the sites with greatest CAL or PD in a participant. However, please note that the mean values of CAL of the surgical sites in both groups were > 7 mm. We added a supplementary Table describing demographic information and clinical parameters of this follow-up study in the revised manuscript.
-The authors state that "calibrated examiners assessed" up to 5 periodontal variables. Don't the authors believe that measures of intra- and inter-examiner concordance agreement should be described?.
(Response)
We thank the reviewer for this comment. We added the information to the revised manuscript.
-With which periodontal probe did the authors conduct the study?.
(Response)
We apologize for the lack of information. We described it in the revised manuscript.
-As the authors describe in the penultimate paragraph of the introduction, they justify the wide use of DBBM in periodontal regeneration, with GTR techniques. Why have the authors not used membranes as a barrier mechanism in their essay, and a fundamental requirement of these techniques?.
(Response)
The design of the original 6-month RCT did not include the use of a membrane.
- How was the regression model built, what were the independent variables?.
(Response)
The regression model was built based on the results of correlation analysis, and the software package Prism was used. As shown in the supplementary tables, the independent variables include baseline parameters such as PPD, number of teeth, BOP, and defect depth.
RESULTS
-The authors continually refer to the two previous published studies, this creates confusion. If the present, I include 25 patients (versus 32) and 32 analyzed sites, (versus 44), therefore if they intend to consider different studies, the authors cannot direct the readers to the previous ones for the demographic information and clinical parameters. The number of patients and the number of sites have changed and readers need to know these facts.
(Response)
In light of the reviewer’s comment, we added the demographic information of the current study in the supplementary table. 
-The authors should under discussion highlight the differences with the two studies already published (at 6 months and at two years) and the justification for evaluating the same variables at 4 years. What does this new assessment contribute? As can be seen in the tables and graphs, most of the clinical and OHRQL variables are identical to those obtained at two years. Only RBF in comparison between groups is different, and when it is compared with the results at 6 months, 1 year, and 2 years, table 2? Why don't the authors provide this information to demonstrate or justify this study?.
(Response)
We thank the reviewer for this important comment. In the Discussion section of our submitted manuscript, we explained the rationale behind conducting this 4-year follow-up study. As already stated above, most RCTs are relatively short term and, due to various reasons, they are seldom re-visited or extended (Davies et al. Scientific Reports, 8(1), 1-8, 2018). Obviously, there is no guarantee that treatment effects remain unchanged beyond the initial or short-time follow-up study. We reported 2-year follow-up results, but 2-year is relatively short follow-up period. Therefore, 4-year evaluation adds to the existing literature by providing longitudinal data on the outcomes of the use of rhFGF-2 with DBBM. In light of the reviewer’s advice, we added such statements in the Discussion section in the revised manuscript.
The fact that the some of the data are not entirely different from the 2-year study is important in its own way, showing the stability of the outcomes.
DISCUSSION
-What does it mean: ….Given that the rhFGF-2 formulation itself lacks space-making property, this finding may further support the clinical efficacy of rhFGF-2 therapy?. As a reader, I take it that since rhFGF-2 is a non-space-creating substance, and since it does not create additional benefits along with DBBM, it would be more logical to use the latter product, which is safe and highly accepted as regenerative.
(Response)
Please note that we are not comparing the use of DBBM alone vs. DBBM+FGF-2 in this study. Although the use of DBBM alone may yield favorable healing, evidence is inconclusive whether the use of bone graft alone can regenerate periodontal tissue.
In this study, the use of rhFGF-2 alone yielded similar general outcomes compared with the combined use of rhFGF-2+DBBM. Considering that rhFGF-2 formulation does not have a space-making property (ex, membrane or scaffold), such clinical outcome can be considered relevant.
In light of the reviewer’s comment, we modified the sentence for clarity.
Round 2
Reviewer 1 Report
The review is correct
Author Response
We are grateful for your time and effort in reviewing our manuscript once again and for the helpful comments and advice.
The review is correct.
(Response)
Thank you for this comment.
Reviewer 3 Report
Three aspects still need to be clarified and/or improved by the authors. They are presented below:
-The authors have entered in M&M that PPD and GR were recorded in 0.5-mm increments. How can you measure 0.5mm increments with a Willians type manual probe with markings at 1,2,3,5,7,8,9,10mm?. For example, a recession of 3.5mm, how can it be determined with this probe? Clarify.
-I insist, the authors say that they have included patients with moderate and severe periodontitis (Stage III periodontitis) and define the inclusion criteria as PPD>4mm and intraosseous defect >3mm. For any reader with clinical periodontal knowledge, these definition criteria do NOT reflect moderate or SEVERE periodontitis. The authors have to clarify these concepts.
-On the question of why they did not use membranes in the RGT technique, it cannot be answered as they do. It is necessary, at least in the discussion, to include some comment that justifies this procedure. Any connoisseur expects the use of membranes in regeneration techniques.
Author Response
We are grateful for your time and effort in reviewing our manuscript once again and for the helpful comments and advice.
-The authors have entered in M&M that PPD and GR were recorded in 0.5-mm increments. How can you measure 0.5mm increments with a Willians type manual probe with markings at 1,2,3,5,7,8,9,10mm?. For example, a recession of 3.5mm, how can it be determined with this probe? Clarify.
(Response)
We understand the reviewer’s concern. Yes, a Williams type probe has markings as you mentioned. We do not feel that the accurate 0.5-mm increment measurement is possible in using such a probe. When a measurement was in between the markings, an estimate was made in 0.5-mm increments.
-I insist, the authors say that they have included patients with moderate and severe periodontitis (Stage III periodontitis) and define the inclusion criteria as PPD>4mm and intraosseous defect >3mm. For any reader with clinical periodontal knowledge, these definition criteria do NOT reflect moderate or SEVERE periodontitis. The authors have to clarify these concepts.
(Resonse)
Thank you for this comment.
This inclusion criteria have been already set in the original study. The criteria of PPD> 4 mm as moderate periodontitis is consistent with the classification of the Clinical Practice Guidelines by the Japanese Society of Periodontology. This information has been added in the previous revision. Because we made patent-based diagnosis (not site-based), the target sites were not necessary the sites with greatest CAL or PD in a participant. However, please note that the mean values of CAL of the surgical sites in both groups were > 7 mm. We added a supplementary Table presenting baseline clinical parameters of this follow-up study in the revised manuscript.
-On the question of why they did not use membranes in the RGT technique, it cannot be answered as they do. It is necessary, at least in the discussion, to include some comment that justifies this procedure. Any connoisseur expects the use of membranes in regeneration techniques.
(Response)
We do not understand what the reviewer meant by “RGT technique”. In this study and two previous studies, we are not evaluating the clinical outcomes of “Guided Tissue Regeneration (GTR)” technique, but the periodontal regenerative therapy, comparing the use of a biological agent with/without a scaffold material. In Japan, the formal approval for the rhFGF-2 formulation (Regroth) in the periodontal regenerative therapy does not include the use of a barrier membrane. In light of the reviewer’s advice, we added comment on this in the Discussion section of the revised manuscript.